# Who is to blame for COVID-19? Examining politicized fear and health behavior through a mixed methods study in the United States

Lisa J. Hardy[1,2]*, Adi Mana[3], Leah Mundell[1,2], Moran Neuman[4], Sharón Benheim[4], Eric Otenyo[2,5]

1 Department of Anthropology, Northern Arizona University, Flagstaff, Arizona, United States of America, 2 Social Science Community Engagement Lab, Northern Arizona University, Flagstaff, Arizona, United States of America, 3 School of Behavioural Science, Peres Academic Center, Rehovot, Israel, 4 Martin-Springer Center for Conflict Studies, Ben-Gurion University of the Negev, Beer Shiva, Israel, 5 Politics and International Affairs, Northern Arizona University, Flagstaff, Arizona, United States of America

* lisa.hardy@nau.edu

**Data Availability Statement:** Raw data cannot be shared according to university IRB and university guidelines due to lack of participant consent for repository use. The university IRB overseeing these

## Abstract

### Background

Political ideologies drove public actions and health behaviors in the first year of the global pandemic. Different ideas about contagion, health behaviors, and the actions of governing bodies impacted the spread of the virus and health and life. Researchers used an immediate, mixed methods design to explore sociocultural responses to the virus and identified differences and similarities in anxiety, fear, blame, and perceptions of nation across political divides.

### Methods

Researchers conducted 60 in-depth, semi-structured interviews and administered over 1,000 questionnaires with people living in the United States. The team analyzed data through an exploratory and confirmatory sequential mixed methods design.

### Results

In the first months of the pandemic interviewees cited economic inequality, untrustworthy corporations and other entities, and the federal government as threats to life and pandemic control. Participants invoked ideas about others to determine blame. Findings reveal heavy associations between lack of safety during a public health crisis and blame of "culture" and government power across the political spectrum.

### Conclusion

Data indicate anxiety across political differences related to ideas of contagion and the maleficence of a powerful elite. Findings on how people understand the nation, politics, and pandemic management contribute to understanding dimensions of health behaviors and underlying connections between anxiety and the uptake of conspiracy theories in public

data does not allow the researchers to share them ethically or legally due to the nature of qualitative research and the lack of participant consent. Please contact the Director of Research Administration, HRPP, Mary Hanabury - 928-523-9428 - Mary. Hanabury@nau.edu with questions.

**Funding:** 1. Northern Arizona Office of the Vice President for Research TRIF Research Acceleration (TRA) program, funded under the State of Arizona Technology and Research Initiative Fund (TRIF), administered by the Arizona Board of Regents. Grant recipient: Dr. Lisa J. Hardy 2. Northern Arizona University College of Social and Behavioral Sciences Research Support Award. Grant recipient: Dr. Lisa J Hardy.

**Competing interests:** The authors have declared that no competing interests exist.

health. The article ends with recommendations drawn from project findings for future pandemic response.

## Introduction

In March of 2020 the United States began to react to the World Health Organization's declaration of the rapid spread of SARS-CoV-2 across the globe. At the time of this writing the pandemic has been raging for over a year with over 4.2 million lives lost worldwide [1] with more than 614,000 of those occurring in the United States [2]. Research emerges every day documenting current and possible future public health concerns related to the virus as well as isolation, economic vulnerability, loss of the opportunity to grieve in community, and other risks to health and life.

The pandemic impacts different populations in unequal ways, with higher rates of virus transmission and adverse events in some populations than others [3, 4]. The context of extreme political and economic inequality predates the pandemic and will outlast this current crisis. Social uprisings, a highly contested election, and public clashes over public health highlight a public focus on inequality [5].

### Political ideology and health during Covid-19

Upon the declaration of a pandemic, people immediately faced anxiety-inducing information and misinformation. Scientists worked tirelessly from the first identification of the virus to understand contagion and manifestations of COVID-19, and journalists hurried to catch up. Political leaders expressed personal opinions and public health directives differently. In the United States, political leaders on two distinct sides took hold of virus information and directed accordingly. Divergent political ideologies resulted in identifiable differences in how people interpreted scientific data and behaved with respect to safety, and these differences impacted rates of virus transmission. Mayors, governors, federal and tribal leaders, and other policymakers began and continue to issue conflicting pandemic response guidelines and engaged in public arguments over present dangers and the correct way to stop the spread of COVID-19 [6].

At the inception of the pandemic, Republican lawmakers were more likely to resist enacting strong prevention measures than Democrats. A narrative from the right included highlighting the importance of a functioning economy as a priority over population health. Leaders from within the Democratic party were generally more likely to follow recommendations from the Centers for Disease Control (CDC) on the priority of public health measures including directives to shelter-in-place, close business, and limit travel. These political ideologies appeared in social interactions through everyday tensions over masks and distancing and, as a result, in measurable health outcomes.

Political differences also appeared in discussions and actions related to the U.S. economy. The focus of Republican commentators, journalists, and the general population on the political right raised economic concerns related to impending loss of personal freedoms contributing to an ideology that mask mandates and business closures were more dangerous than the virus. In interviews, people discussed mask mandates as "muzzles" and part of a plan to control the general public.

People on the political left expressed more concern for public health. In June of 2020 Republicans were also more likely to believe the pandemic was nearly over than Democrats

who remained highly concerned [7]. Many right-leaning people living in the United States focused on economic viability, whereas left-leaning people were more likely to adhere to the recommendations of White House Medical Advisor and Director of the U.S. National Institute of Allergy and Infectious Diseases, Dr. Anthony Fauci, and the World Health Organization (WHO). Dr. Fauci served as medical director under the then-Trump administration; however, many supporters of Trump were suspicious of him. Over time there were changes on beliefs about economic concerns, personal freedom, and health mandates tough the political divide continues to remain paramount in shaping U.S. perceptions of COVID-19.

## Impacts on health response

In this article, we trace different and similar pandemic epistemologies, arguing that the root cause of many national anxieties is the accumulation of wealth and power, though the end-points of these logical pathways diverge. Participants across the political spectrum articulated fears and anxieties about elites and distant institutions controlling their lives with maleficent intent such as depopulating the world for economic gain. Findings represent a convergence of popular opinions across the political spectrum.

Throughout history, epidemics have become loci of xenophobia and blame. This is also true during this pandemic when anti-Asian hate crimes have risen around the world. Many people understand the pandemic as a blameless, natural part of life, though many also blame others for poor management of the crisis. Many use the pandemic as justification for xenophobic responses. Hate crimes represent a present health crisis that accompanies the pandemic in addition to direct impacts of the virus.

Scholars and health professionals also call for investigations of the role of capitalism in the current crisis [8]. These findings contribute to the investigation of the role of political and economic power and difference in pandemics. They also inform research on mistrust of public health directives and the implementation of prevention and pandemic control as the pandemic exposes inequalities of a healthcare system that does not consider health and healthcare to be a human right [9].

## The study

This study began in the first weeks of the pandemic. A researcher quickly designed the first phase of an exploratory study inquiring about perceptions and impacts of the pandemic. Upon approval from the Institutional Review Board, the researcher began phone interviews. Within several months, the project grew to include a national team and partnerships with international social science researchers also responding quickly to the pandemic. The team used an inductive and iterative approach to refine interview materials and account for change over time. The international group developed a questionnaire that researchers disseminated multiple times in nine countries [6]. This article draws on an analysis of 60 interviews conducted between March of 2020 and March of 2021 and 637 of over 1,000 questionnaires collected in a second round of data collection in early 2021. While the study is not specifically about perceptions of political difference and partisanship, it became quickly relevant. In interviews and questionnaires, we asked people to self-select their political affiliation and used these to analyze data. Self-defined right-leaning participants in this research were more likely to be Republican or Independent voters and self-defined left-leaning people were more likely to be Democrats who define themselves as liberal or left-of-center. In this article we draw on self-identified partisan affiliation to examine differences and similarities in left-leaning, center, non-political, and right-leaning groups, differences that are significant across data sets.

## Theory

This is a mixed theory and mixed methods study drawing on two medical social science perspectives that we used to design and analyze data, Critical Medical Anthropology (CMA) and salutogenesis arising from medical sociology. CMA provides a political economic lens through which to evaluate and understand where and how political economic power and inequality inform and imbue public perceptions and actions of bodies, illness, and–in this case–a virus. CMA incorporates syndemics to trace and investigate multiple axes of inequality impacting differences in health and life [10, 11]. This is not to be confused with the current debate over the use of the term syndemic to define COVID-19 globally, which Mendenhall argues erases locally specific and historic roots of racism and political inequality [12, 13]. Instead, the term can be used to describe multiple, overlapping inequalities enmeshed within particular histories and presents of inequality and racism, which generate unequal outcomes. CMA also allows for inquiry regarding on-the-ground, lived experiences [14] and meaning made through relationships and experiences, and, at the same time, a critical view of larger political and economic systems. Further, we drew from scholarship connecting national and global political-economic moments to perceptions of virus and disease [15] and economic theory of political change through hegemonic neoliberalism [16].

The concept of neoliberalism, while not all encompassing, is useful in our analysis and interpretation of pandemic data in the United States. Neoliberalism is defined as an economic system based on individuals, private property, and accumulation in contrast to political economic plans characterized by investment in safety nets and community good [17]. Under neoliberalism, public perceptions of suffering become characterized as personal failures rather than outcomes of structural inequality [18]. It is a hegemonic social and political system that is defined by "extensive accumulation of interest-bearing capital" [18]. This plays out during pandemic times in different ways including the valorization of frontline heroes, which can, in practice, "mask neoliberal violence," [19] and "free market" healthcare systems designed to increase accessibility for those in advantageous positions. These concepts are easily understood as the accumulation of wealth by a few powerful individuals and institutions and the idea that anyone can pull themselves up by their bootstraps if they try hard enough.

We also designed research instruments to explore salutogenesis [20] with a focus on coping resources that enable people to manage crises and remain healthy. Salutogenesis is ". . . a global orientation that expresses the extent to which one has a pervasive, enduring though dynamic feeling of confidence that one's internal and external environments are predictable and that there is a high probability that things will work out as well as can reasonably be expected" [20]. There are three main aspects of this concept that cover comprehensibility, manageability, and meaningfulness [21].

Salutogenesis suggests that people have varying levels of sense of coherence (SOC), expressing an individual view of the world as comprehensible, manageable, and meaningful. In the current study we were interested in exploring individuals' perceptions of their coping resources in a national context. This framework includes a focus on a process through which individuals find health through coherence when making sense of the world [22]. The pandemic presents the challenges of chaos that people manage through the invocation of manageability and the use of resources spanning from the individual to family to larger society. Recently, researchers have extended the concept of salutogenesis to encompass how people make sense of political systems and national ideologies [23].

We were mainly interested in sense of national coherence (SONC) and trust in governmental institutions as helping to reduce levels of anxiety during the outbreak of the pandemic. SONC integrates three components central to SOC into a national level of comprehensibility, manageability, and meaningfulness. Comprehensibility is the perception that one's life in a

particular national context is predictable, safe, and secure and that the actions of one's group are comprehensible and logical. Manageability is the perception that people are safe and cared for by a nation that is available to assist and meet needs and demands. Manageability is also related to confidence, trust, and satisfaction in the nation's institutions and systems and to the perception of the nation's efficacy and readiness to cope in times of stress and danger. Meaningfulness is the perception that one's national group is meaningful because it provides challenges, goals, vision, and shared aspirations. Recent studies in the period of COVID-19 revealed that SONC reduced anxiety among people from different countries [6] However, higher levels of SONC were also related to right-leaning voting patterns.

This article focuses on only the United States data from an international project. CMA and salutogenesis complement each other in analyzing experiences of the pandemic occurring within a politically divided and highly unequal society in this national context. Over the past decades, economic inequality has continued to grow in the United States, as stagnating wages for workers and reduced taxation of the wealthy have led to a shrinking middle class. As people experience the loss of safety net resources and decreasing life expectancy in the United States, they have struggled to make sense of the political and economic national contexts. Crushing inequality is not new in the United States, founded on colonial theft and waves of slavery. Many have pointed out that the inequality of COVID-19 contributed to a moment of disruption that challenged the longstanding national narrative of the United States as a free and equal society.

## Methods

We report here on one section of an ongoing mixed methods study including 60 qualitative semi-structured interviews and 637 survey responses, all from people living in the United States 18 years and older. In each data collection method, we used several sampling strategies to obtain a wide range of views from people with varying genders, ethnic identities, ages, and political and religious affiliations.

### Data collection and analysis

The team used a sequential, mixed methods research design for data collection and analysis [24]. We began with inductive, qualitative inquiry and drew on early analysis to develop and select several open-ended questions for the end of a survey. We used mixed methods analysis design to develop codes and apply them to responses to the open-ended questions and interview transcripts for a triangulation of findings. Exploratory survey analysis led to iterative data collection and analysis within interviews. We had two teams of researchers working together. One team contributed expertise in qualitative methods and analysis and the other drew on their expertise in quantitative statistical analysis. Both teams met regularly to discuss findings and explore both data sets.

The team triangulated themes through this mixed methods approach. This strategy represents methods, investigator, and theory triangulation [25–27]. Method triangulation is used to compare and analyze each data set alone and both data sets together [28]. This design also allowed us to corroborate across theories and investigators to enhance overall findings and investigate patterns through multiple means [25]. The team designed the iterative and sequential aspect of the design invoked triangulation at punctuated intervals as depicted in Fig 1.

### Interviews

We began interviews at the beginning of the pandemic prior to the existence of research on what was to come. We responded to the lack of knowledge and existing confusion through the development of an exploratory, inductive, semi-structured conversational style interview conducted over the phone with participant volunteers [29]. This strategy allowed us to iteratively

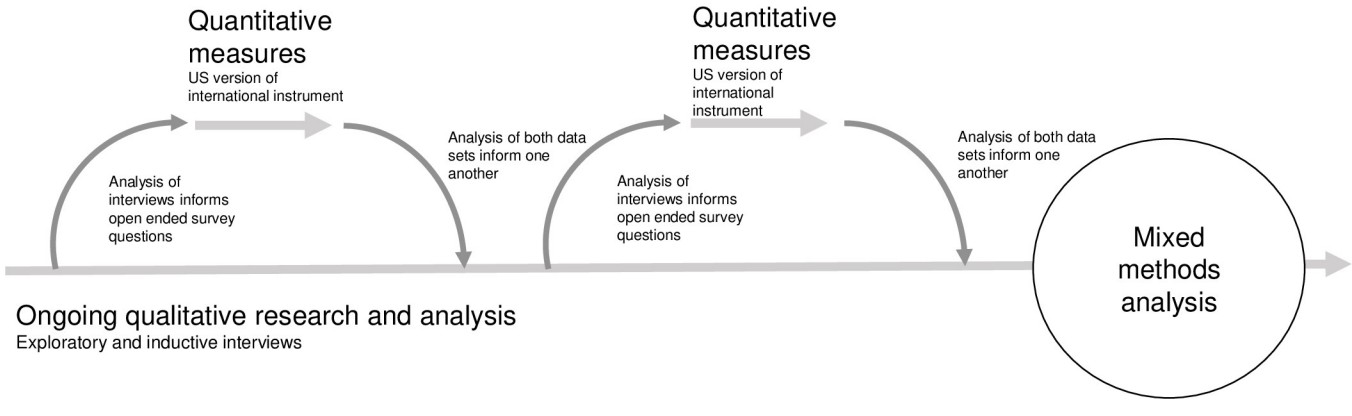

**Fig 1. Mixed methods analysis.**

develop theory while building on existing literature [30, 31]. Interview were 1–2 hours in length, recorded, and fully transcribed. Inquiry ranged from general experiences and perceptions of COVID-19 and the pandemic, to coping strategies, fears, and other related topics (see S1 Appendix for full interview guide). Interviewers created memos when possible for ongoing discussion and analysis.

Interview recruitment began with a snowball sampling technique and continued into a nominated expert strategy [32]. Interviewees reported enjoyment of the process and were enthusiastic about reaching out to contacts who would then email us with interest. We selected interviewees based on a demographic screener. The screening instrument allowed us to monitor who was included and to select targeted volunteers from the available pool when necessary. When we noted that we had interviewed significantly more women, for example, we selected men and gender non-conforming volunteers for interviews. Our goal was not to have a fully representative sample for the number of interviews but to avoid an extremely limited sample. We made minor iterative changes in the interview guide to account for change over time and were careful to continue consistent questions to understand saturation and consistency. We also collected volunteer information in the second questionnaire, allowing us to expand our pool of potential interviewees far beyond any existing networks.

## Questionnaires

This project includes a partnership with nine countries disseminating a similar questionnaire in different national contexts. Data for this article came from the U.S. version of the questionnaire with design and analysis originating with the international team. We disseminated the questionnaire twice in the first year of the pandemic. In this article we include survey data collected after the election in November 2020 and included 637 participants. For quantitative data we used SPSS software, and the analyses include descriptive statistics, Alpha Cronbach correlations and ANCOVA comparison between groups of voting affiliations (far right, right, center, left, far left, and non-political). The questionnaire (provided in S2 Appendix) includes the following measures, some of which were developed by the international team, and several open-ended questions related to the ongoing analysis of interviews.

## Research tools

**Generalized Anxiety Disorder (GAD-7) [33].**  The 7 items of this scale inquire about the degree to which the participant has been bothered by feeling nervous, anxious, worried,

restless, annoyed, and afraid during the 2 weeks prior to answering the questionnaire. Each item was scored on a 4-point Likert scale (0–3), with total scores ranging from 0 to 21 where higher scores reflect greater severity of anxiety. Internal consistency of the questionnaire was estimated at 0.89 [26] and in the current study α = 0.94, 0.94, 0.90, 0.90, 0.90, 0.92 among far right, right, center, left, far left, and non-political accordingly.

**Sense of National Coherence (SONC) [6].**    The 8 items on a 7-point Likert scale (1 = totally agree, 7 = totally disagree) explore the participants' perceptions of their own society as comprehensible, meaningful, and manageable. Internal consistency of the questionnaire was estimated at 0.80 [6] and in the current study α = 0.67, 0.70, 0.73, 0.66, 0.80, 0.74. among far-right, right, center, left, far-left and non-political accordingly.

**Trust in governmental and other institutions.**    A 13-item questionnaire examines level of trust in relevant institutions (i.e., media, legal courts, police, the incoming president, the outgoing president, the government, the economy, the CDC, doctors and other healthcare workers, schools, state government/governor, local city or town leaders, religious leaders/leadership, scientists, pharmaceutical companies) on a 5-point Likert scale (1 = very much, 5 = not at all). Internal consistency was α = 0.91, 0.87, 0.85, 0.76, 0.88, 0.95, among far-right, right, center, left, far-left and non-political accordingly.

**COVID-19 health behaviors.**    Two items explore health behaviors related to the most conflictual COVID-19 directives: wearing a mask and vaccination. The first related to health behaviors of mask wearing, on a 7-point Likert scale (1 = totally agree, 7 = totally disagree). The second item explored openness to receiving the vaccine by asking "Do you plan to receive the Covid vaccine?" (1 = yes, I already had the vaccine, 2 = yes, I will receive the vaccine as soon as it becomes available to me, 3 = probably 4 = I don't know yet/haven't decided, 5 = not likely, 6 = definitely not.)

**Demographic variables.**    We asked about the participants' age, gender, ethnicity, household income, and religious affiliation. We explored perceptions of health risk by asking if the participant was part of a risk group because of age and/or medical condition.

The questionnaire also included several open-ended questions that reflected areas of inquiry from data in the interviews. For example, we ask the question "who is to blame for the COVID-19 pandemic?" While this may seem like an odd, leading question, it emerged from qualitative interviews as an important area for inquiry to understand blame and mistrust among people with different political orientations. It also reflected early conversations about the pandemic in public forums where questions about cause and blame were paramount to rapidly unfolding information and misinformation.

**Recruitment.**    Survey recruitment included the use of social media and personal networks at first and the use of the Prolific service for around 15% of respondents to round out demographic variables. We aimed for a balance of gender, ethnicity, and political orientation and also collected information on sexuality, age, household income, number of household members, state of residence, and religious affiliation.

## Analysis

Our mixed methods analysis design, pictured in Fig 1, allowed for the exploration and analysis of themes based on multiple vantage points. We used qualitative and quantitative methods and analysis to understand and document each theme included in this article. Analysis of qualitative data included a mixed methods approach to identify thematic consistency and changes over time in both data sets through the development of a coding frame [34]. To analyze interviews and open-ended responses, we checked for salient themes through the development of three independent code lists by three different coders. We added these codes to a table and

prioritized those that appeared in two of the three categories. We deleted any codes that appeared in only one category.

We found a high level of consistency among coders and, through our systematic approach of identifying and applying codes, removing outlying codes, and working together to discuss any discrepancies. We achieved intercoder consistency, a proven strategy for qualitative analysis [34, 35]. This check for intercoder consistency was enhanced by the use of three coders who participated as researchers on the project and had extensive methods training. This group had not discussed interview data at length or shared insights on findings prior to coding the interviews and only entered a more collaborative analysis phase once researchers decided on agreed-upon themes and codes.

Our data collection and analysis design for qualitative data included ongoing analysis through memos and theme identification. At the conclusion of the first 30 interviews, we conducted analysis by through development of a coding frame and codebook. For analysis on this specific content, we broke the interviews into sets of 20, organized sequentially, and applied related codes to each transcript allowing us to identify consistency and change over time. We selected sections of all transcripts that included content about trust, blame, and perceptions of pandemic response for this analysis and developed thematic consistency through the application of coding between three researchers. We deleted outlier codes and discussed any discrepancies that appeared to belong in a single category to resolve these coding differences. Once we applied codes to survey and interview data we checked for the weight and consistency of answers in batches over time to identify consistency and change. We then focused on several salient identified themes for a deeper analysis and triangulated the questionnaire and interview data to understand these trends in more depth.

For the questionnaire analysis, we applied analysis to the instruments included in the international survey and focused on political leaning, a theme that remained consistent in interviews and the questionnaires as a significant difference for people living through the pandemic in the United States. We divided the participants into six groups according to how they answered a question about political leaning and another about party affiliation. 22 participants defined themselves as far right, with an age range of 24–64 years (mean = 38.1, SD = 9.3); 114 participants defined themselves as right, with an age range of 20–82 years (mean = 39.71, SD = 15.32); 154 participants defined themselves as center, with an age range of 18–77 years (mean = 39.37, SD = 15.97); 163 participants defined themselves as left, with an age range of 19–75 years (mean = 41.06, SD = 17.22), 88 participants defined themselves as far left, with an age range of 19–79 years (mean = 37.3, SD = 15.67); and 111 participants defined themselves as nonpolitical, with an age range of 18–84 years (mean = 33.65, SD = 11.84) (see Table 1).

All research and analysis processes are covered under an approved university Institutional Review Board protocol.

## Results

Results of mixed methods data analysis indicate, not surprisingly, differences across political affiliation. In the below sections, we discuss patterns in the national coping resources of sense of national coherence, trust, health behaviors, blame, and anxiety across groups as well as cultural essentialism and blame for the virus, perceptions of late capitalism and economic inequality, and relationship to the US government. These are salient themes identified through inductive qualitative coding of interviews and open-ended questions that we further examined through quantitative analysis. We begin with results from the questionnaire and move into the layered analysis of both data sets to elucidate patterns and themes from this mixed methods research.

**Table 1. Demographic variables: Gender and belonging to a risk group.**

| Variable | Political Affiliation | | N (%) |
|---|---|---|---|
| Gender | Far-right | Male | 13 (59.1) |
| | | Female | 8 (36.4) |
| | | Non-binary | 1 (4.5) |
| | | Total | 22 |
| | Right | Male | 62 (54.4) |
| | | Female | 49 (43) |
| | | Non-binary | 3 (2.6) |
| | | Total | 114 |
| | Center | Male | 58 (37.7) |
| | | Female | 91 (59.1) |
| | | Non-binary | 5 (3.2) |
| | | Total | 154 |
| | Left | Male | 49 (30.1) |
| | | Female | 111 (68.1) |
| | | Non-binary | 3 (1.8) |
| | | Total | 163 |
| | Far-left | Male | 24 (27.3) |
| | | Female | 58 (65.9) |
| | | Non-binary | 6 (6.8) |
| | | Total | 88 |
| | Non-political | Male | 55 (49.5) |
| | | Female | 55 (49.5) |
| | | Non-binary | 1 (0.9) |
| | | Total | 111 |
| Belonging to risk group due to age or health status | Far-right | Yes | 4 (19) |
| | | No | 17 (81) |
| | Right | Yes | 39 (34.2) |
| | | No | 75 (65.8) |
| | Center | Yes | 60 (39) |
| | | No | 94 (61) |
| | Left | Yes | 63 (39.6) |
| | | No | 96 (60.4) |
| | Far-left | Yes | 29 (33.7) |
| | | No | 57 (66.3) |
| | Non-political | Yes | 31 (28.4) |
| | | No | 78 (71.6) |

Interview participants across the political spectrum indicated high levels of mistrust of other people, larger entities, and governments in terms of suggesting other entities are to blame for either the pandemic or problems related to the management of the pandemic. People across the spectrum reported anxiety about the federal government and overall mishandling of pandemic response. The exception to this occurred in highly religious participants who communicated a trust that they would be taken care of and that "God is in charge." Other than these interview responses there was a high level of consistency across interviews in the lack of trust; however, the specific entities and government institutions who were not trusted varied by political affiliation.

Participants on the political left expressed mistrust of former president Donald Trump, political leaders, and large corporations. This included pharmaceutical companies. There were higher levels of trust in scientists on the left. Economic inequality and capitalism also appeared throughout interviews and survey responses as problems that laid the foundation for public health disaster prior to and during the pandemic. On the far left several outliers discussed political plots for depopulation of the world, with two talking about being "boiled like frogs in a slowly heated pot." People on the left were also more likely in interviews to report having been victims of hate crimes and public attacks during the pandemic. In later interviews, people told several stories of aggressively racist interactions they endured in public places.

On the political right there were also high levels of mistrust. These interviews included suspicion toward individuals, entities, and governments as well. The individuals who were not to be trusted included people who believed everything they were told, called "sheeple" by some. Entities that this group mistrusted included the World Health Organization, Centers for Disease Control, and global governing bodies. Like those on the far left, people on the far right also discussed the pandemic as a strategy to depopulate the world. Notably, the Chinese government and cultural practices in China were listed often as recipients of blame among the political right.

For some there appeared to be a sense of an undefined "they" who are intentionally causing disaster and intentionally leaving some people out of the plan for survival. One person said "...They are still trying to scare people. There are people that [are] going to be scared. We haven't seen the end of the economic impacts of the disease. At the same time, I feel like the big things are not of dying from COVID-19 but of people dying from other things." The risk of death from impacts of the pandemic extended to mistrust of the medical system. A medical provider said, "There are people that I knew that needed medical treatments but were refused because it wasn't COVID-19. I had an acquaintance who just discovered she had cancer and they didn't do anything about it because it wasn't COVID-19."

When we explored political differences in trust, health behaviors and sense of national coherence (SONC), questionnaire responses revealed some similar findings and some fine grained and important differences. ANCOVA tested the differences in the levels of the research variables (anxiety, SONC, and trust) between the six research groups (far right, right, center, left, far left, non-political). Gender, age and belonging to a health risk group were entered as covariate (see Table 2).

The results indicated significant differences between voting groups in levels of SONC. Games-Howell post hoc tests revealed that the levels of SONC among right-leaning participants was significantly the highest ($p < .001$) compared to the other groups. SONC among far right and right-leaning participants was higher [$F(5,627) = 17.46$, $p < .01$] compared to the levels of SONC among left and far left participants. This finding confirmed the assumption that the coping resource of perceiving the nation as a place that is meaningful and manageable is more available to right leaning voters than to left-leaning voters. The levels of anxiety were also significantly different between the groups [$F(5, 635) = 3.49$, $p < .005$]. The levels of anxiety among the far left were the highest ($p < .001$). In addition, significant differences were found between far left and left compare to levels of anxiety among right and far right-leaning participants which were lower. This finding also supports qualitative findings that facing the national crisis is more threatening for people in the opposition.

The study also explores the group differences in levels of trust in federal government, and in levels of health behavior and beliefs about COVID-19 regulations. As can be seen from Table 2, no significant differences were found between the groups [$F(5,627) = 1.93$, $P = .088$, *n. s.*] in levels of trust in the federal government institutions who are aiming to control the pandemic. However, levels of self-reported health behaviors such as mask-wearing [$F = (5,621) = 10.34$, $P < .0001$] and willingness to be vaccinated [$F = (5,619) = 17.59$, $p < .0001$] were higher

**Table 2. Means, standard deviations, and ANCOVA analyses.**

| Variable | Group | Mean | STD |
|---|---|---|---|
| SONC | far-right | 4.26 | 0.20 |
| | right | 4.38 | 0.09 |
| | center | 4.12 | 0.07 |
| | left | 3.70 | 0.07 |
| | far-left | 3.30 | 0.09 |
| | non-political | 4.00 | 0.09 |
| Trust | far-right | 3.13 | 0.15 |
| | right | 2.93 | 0.06 |
| | center | 3.08 | 0.06 |
| | left | 3.16 | 0.05 |
| | far-left | 2.96 | 0.07 |
| | non-political | 3.08 | 0.07 |
| Anxiety | far-right | 8.39 | 1.14 |
| | right | 8.23 | 0.51 |
| | center | 9.05 | 0.44 |
| | left | 9.78 | 0.42 |
| | far-left | 11.20 | 0.58 |
| | non-political | 8.21 | 0.51 |
| Self wear mask | far-right | 3.91 | 0.19 |
| | right | 4.14 | 0.08 |
| | center | 4.25 | 0.07 |
| | left | 4.73 | 0.07 |
| | far-left | 4.76 | 0.10 |
| | non-political | 4.37 | 0.09 |
| Others wear mask | far-right | 3.43 | 0.20 |
| | right | 3.39 | 0.09 |
| | center | 3.23 | 0.07 |
| | left | 3.10 | 0.10 |
| | far-left | 3.10 | 0.09 |
| | non-political | 3.30 | 0.09 |
| Vaccine | far-right | 4.38 | 0.26 |
| | right | 3.84 | 0.11 |
| | center | 4.16 | 0.10 |
| | left | 4.92 | 0.10 |
| | far-left | 4.87 | 0.13 |
| | non-political | 3.93 | 0.11 |

Sense of national coherence, trust, anxiety and compliance (self mask-wearing, perception that others comply with mask-wearing, willingness to be vaccinated) among voting groups. Covariates: Age, gender, belonging to a risk group.

among far left and left-leaning participants compared to far right and right participants. We did not find differences between the groups in perceiving the others as complying with COVID-19 regulations and wearing masks [F = (5,612) = 1.90, p = 0.09, n.s.]. The results revealed that while far right and right-leaning participants reported higher levels of SONC and lower levels of anxiety compared to the far left and the left-leaning participants, the latter reported greater likelihood of adhering to COVID-19 regulations and health behaviors.

We conducted Pearson correlations separately among each of the six research groups to explore the relationships between the research variables and revealed that among far right, left, and far left, strong relationships were found between SONC and trust (r = .65, .64, .76, p < .001, accordingly). Positive and weaker correlations were found between SONC and trust among right, center, and non-political (r = .23, .33, .13, p < .05). It seems that among all groups SONC is related to trust in authorities responsible for controlling the pandemic.

The relationships between anxiety and SONC were negative and significant only among the right, center, left and far left-leaning participants (-.23, -.15, -.31, -.26, p < .05).

The correlations between anxiety and trust were negative and significant among left and far left-leaning voters (r = -.28, -.24, p < .05), while among the right-leaning participants we identified positive correlations between anxiety and trust (r = .28, p < .05). It seems that SONC and trust are related to lower levels of anxiety; however, these relationships are weak and inconsistent.

While exploring the relationships between SONC, trust, and health behaviors of self and perceived others as wearing masks, the results revealed that for far right and far left participants, strong correlations were found between perceiving others as wearing masks, with SONC (r = .67, .50, p < .001) and trust (r = .65, .53), while self-wearing mask was not related to SONC and trust among those groups. We identified a similar direction but weaker correlations between perceptions of others and SONC and trust among left leaning voters (r = .19, .29, p < .05).

It seems that perceiving others as following health directives is an important factor related to SONC and trust.

## Trust

One of the most prominent areas of trust and mistrust that arose in interview data was blame for the pandemic related to the U.S. government. This relationship between trust and SONC reveals varying levels of mistrust among different groups that is similar across political lines even when the reason for the mistrust is different. Discussions of perceptions of the federal government spanned from numerous responses about then-president Trump as the culpable individual to the whole administration of the federal government. One left-leaning interviewee said:

> The U.S., of all countries, has chosen the worst path for response to COVID. Inconsistent, fragmented, ineffective. We have failed to control disease spread by refusing to do what is necessary to control it, while also refusing to provide appropriate economic relief to those who need it desperately. The incoming President and the vaccine bring hope for a better future, but we cannot bring people back from the dead or revive the economy overnight. For the U.S., recovery will be slow and difficult. The outgoing President has sown division, death and instability. It did not have to be this way.

On the political right there was also mistrust and blame associated with the government enmeshed in the idea that liberal politicians were inflating the dangers of the virus to restrict individual freedoms. In this group there was a higher level of SONC even when mistrust of the government prevailed as a factor in pandemic anxiety. One self-described "right of right" interviewee said:

> [The pandemic] has been a *great* opportunity a *fantastic* opportunity for the government to see who is compliant and who's not. I mean viruses come and go, the difference between

this one and all of them previous to it is they utilized this one to *absolutely instill fear in the public* and *I'm talking on a global scale.*

These quotes are both illustrative of patterns we identified across interviews. Mistrust of the federal government existed across the political spectrum, though the reasons for the mistrust diverged. There was also mistrust of the way the pandemic became politicized. One participant echoed the words of many others when she said of the government's pandemic response, "I think that the strategy is more about a political strategy than a health strategy. Instead of public health."

Trust also varied by partisanship. Right-leaning participants were more likely to trust the economy, police, Trump, and religious leaders, while left-leaning participants were more likely to trust media, Biden, the government, legal courts, pharmaceutical companies, scientists, the Center for Disease Control (CDC), and healthcare workers. This difference in levels of trust also adheres with a higher sense of national coherence on the right, indicating a sense of nationalism that relies on military-style leaders. The increased trust on the right is also coupled with qualitative data that shows right-leaning beliefs about mistrust are related to ideas of economic conspiracy. Trust in "the economy" is possibly related to an idea that it is a natural function while people and entities who accumulate extreme wealth are included in entities that are objects of mistrust like the CDC.

## Blame

We explored both data sets to understand perceptions of blame for the pandemic. In the first months of the pandemic, blame became an important part of the public conversation and one that showed strong splits along party lines. This is one aspect of the pandemic that shows a divergence between political parties in the United States.

Overarching categories included mistrust in a powerful entity, including the World Health Organization and the U.S. or Chinese governments. Other common and related themes were general mistrust of inequality, with the idea that some people or some systems were making decisions and acting in ways that were untrustworthy and unsafe for the general public.

Participants who identified as right-leaning or non-political were more likely to state that Chinese people, the Chinese government, or Chinese markets ("eating the bat") were to blame for the pandemic, while left-leaning respondents were more likely to point toward capitalism and economic inequality; however, there was also consistency across groups.

Results of inductive interview data and open-ended replies on the questionnaire suggested correlations between concerns about governance and management of the pandemic and mistrust of global elites. Our team independently developed codes from the open ended question "who is to blame for the pandemic?" in interviews and questionnaire responses and developed the code list using the process described in analysis. Coded replies to questions about who is to blame for the pandemic included companies/entities (WHO/CDC), capitalism/inequality, government/government leaders, Trump, the virus, China/Chinese government, consuming animals/meat markets, lack of preparedness, behavior of others, ignorance/misinformation, no one, everyone, and other.

The quantitative team explored the question "who is to blame for COVID-19? on the second survey (conducted in the weeks following the presidential elections) using T-tests that revealed significant differences between the research groups in blaming Trump $[\chi^2(5,611 = 35.37, p < .0001)]$, the government $[\chi^2(5,611 = 28.62, p < .0001)]$, China $[\chi^2(5,611 = 110, p = .000)]$, natural forces $[\chi^2(3,503 = 11.03, p = .03)]$, and not blaming anyone $[\chi^2(5,503 = 18.43, p < .0001)]$. (See Table 3).

**Table 3. Frequencies and percentages for the most common responses to the question "who is to blame for COVID-19?" among voting groups.**

| Group | China | Government | Trump | Nature | Others' behavior | No one |
|---|---|---|---|---|---|---|
| **Far right** | 10 | 2 | 0 | 1 | 3 | 0 |
| | (45.5%) | (9.1%) | (0.0%) | (4.5%) | (13.3%) | (0.0%) |
| **Right** | 55 | 3 | 4 | 2 | 1 | 13 |
| | (50.9%) | (2.8%) | (3.7%) | (1.9%) | (0.9%) | (12.0%) |
| **Center** | 32 | 14 | 11 | 6 | 11 | 22 |
| | (22.9%) | (10.0%) | (7.9%) | (4.3%) | (7.9%) | (15.7%) |
| **Left** | 14 | 33 | 32 | 4 | 9 | 32 |
| | (9.5%) | (22.3%) | (21.6%) | (2.7%) | (6.1%) | (21.6%) |
| **Far left** | 2 | 21 | 21 | 5 | 10 | 20 |
| | (2.3%) | (24.4%) | (22.4%) | (5.8%) | (11.6%) | (23.3%) |
| **Non-political** | 9 | 17 | 8 | 11 | 7 | 8 |
| | (8.4%) | (15.9%) | (7.5%) | (10.3%)* | (6.5%) | (7.5%) |
| **Total** | 110 | 67 | 55 | 23 | 28 | 75 |
| | (21.9%) | (13.3%) | (10.9%) | (4.6%) | (5.6%) | (14.9%) |

Far left and left-leaning participants tended more to blame the government and Trump and tended less to blame China, compared to the far right and the right-leaning participants. However, they were also more likely to claim that there is no one to blame. People who defined themselves as non-political were significantly higher, as compared to the others, in blaming natural forces for the crisis. Men also blamed China more often than women ($\chi^2(2,612 = 6.44$, p = .04), and women more often blamed Trump ($\chi^2(2,612 = 12.51$, p = .002). Participants who belonged to a health risk group also tended more than the other participants to blame Trump ($\chi^2(1,603) = 8.19$, p = .004).

People across the political spectrum either stated confidently or questioned out loud associations between cultural practices and contagion. This most often took the form of beginning with the acknowledgement that the first reports of Coronavirus came from China and quickly leaping to blaming China, Chinese people, and Chinese practices. This form of cultural essentialist xenophobia was not limited to China, however. People also discussed other human behaviors in ways that indicated essentialist readings of human behavior among people with less power or in disadvantageous positions.

Discussion of China, the Chinese government, and Chinese cultural practices were common across the political spectrum; however, among participants who identified themselves as right, libertarian, or center, China appeared more frequently as a locus of blame for the pandemic. Some listed only the word China as an entity to blame. Some people who talked about China in interviews did so with verbalized curiosity rather than confident conclusions by querying about military and biological experiments and following that up with "time will tell," or "I really don't know."

Others specified the Chinese government or military and others talked about food markets in China. In interviews and surveys, the idea that there are widespread cultural practices related to wet markets and the eating of bats and other creatures unfamiliar to the US appeared throughout, with one survey respondent answering, "who is to blame" with "the guy that ate the bat?" Some people discussed the consumption of animals in China as something that they found to be repulsive, yet they couched this in an assertion of cultural understanding or an overt discussion of why it would be wrong to assert blame. This left-leaning interviewee responded in a way that echoed throughout other interviews when responding to questions about blame.

> I don't think anyone's to blame for it. As far as the disease making the jump into humans, I mean, no one's to blame. . .I'm very empathetic to other cultures and cultural relativism. Just because people do things differently in other countries does not make them bad or less evolved and we shouldn't discriminate against any other cultures for things like that.

In interviews people reflected on this question from across the political spectrum.

> I'm not a scientist. I don't have a science degree, but the fact of the matter is that the people eating raw bats or slightly cooked bats or whatever they were doing I think it's weird that all the sudden we have this virus and I guess it wiped out this cult in China up on a hilltop somewhere.

Still, the blame for the pandemic returned to China, a xenophobic response to the global catastrophe. The agent of blame in this answer appeared to be government actions removed from Chinese people, untrustworthy science conducted in untrustworthy labs who leaked the virus either intentionally or unintentionally, and abhorrent (to the respondent) cultural practices that led to disaster.

In addition to blame of China and Chinese people, some also blamed other groups and cultural practices. In a discussion of contagion, one white, left-leaning woman living in the Southwest said about Indigenous people:

> . . .they have been a little slower to adopt the precaution measures. I understand. I know many Native Americans they have a communal culture, and I don't know? how they do their thinking. They are living closer to each other because they are poor, many are poor, that's probably the reason and it's not natural for them to keep distance. It is easier for them to forget to protect themselves and that's why they are more exposed but then they are exposing others.

This echoed other responses in that the speaker both blamed "culture" and also people for "exposing others." This statement is not only reflective of contagion blame; it also does not reflect the situation in the area. In fact, local tribes were much quicker and more effective at enacting protections than states. Contagion on tribal lands was also related to variable factors including the need to move across borders and boundaries to do essential work for income on which families survive. This and other examples from interviews indicate instead an idea that blame can be assigned to cultural practices.

The use of the logic of racism to blame people for contagion was also described by people who encountered it themselves. A self-described white, Hispanic woman explained having an unmasked white man approach her in public during the height of the pandemic and tell her he wished she "would die" and that she should take her children back to Mexico.

## Accumulation of wealth and power

There was a general agreement that COVID-19 did not impact everyone in the same way and that COVID-19 had exposed existing inequalities to people who were previously able to ignore them. Interviewees made connections between the disparate outcomes of the virus and co-occurring social movements. Many people in both interviews and surveys reflected on concerns that more people would die if they were poor or suffering from health disparities. Many also said they blamed the US and other governments for not responding in ways that would support equal outcomes:

> I think things such as employment and access to capital have been eliminated. Who are the essential workers and how much do they make? If they are so essential why are you paying them minimum wage? . . .I am glad all of that is being brought to light [it's] not new to me but I like that it's being raised. . .hopefully something will be done about it. More money invested. More awareness in terms of. . . like I do think COVID between COVID and Black Lives Matter and the "defund the police" they are connected and separate in how they unfolded. . .

This notion that the inequality of the pandemic is not new was one that others echoed as well. There was some consensus that the outcomes of COVID-19 would reveal rather than cause existing rifts. This interviewee discussed how the "havoc" of the pandemic "exacerbated" the inequalities but did not create them.

> [The pandemic] wreaked havoc on the entire population and there were a lot of people that died. A lot of people were sick. It shut down every system, restaurants, schools, stores and it exacerbated a lot of the inequities in the system and education system, it polarized people politically. . .

This awareness and focus on inequality also translated into a mistrust of powerful entities including "the media" as well as financially driven healthcare entities and government bodies. Across the political spectrum, respondents distinguished between what they saw as independent scientists and scientists who might have an economic interest in large, multinational businesses. When asked if they trusted different entities such as scientists and doctors, some interviewees responded, "it depends who's paying them." The same was true for the World Health Organization and the Centers for Disease Control. Some reported a trust in these institutions and others suggested that they had been corrupted by powerful leaders interested in population control or financial gain.

Mistrust related to inequality also extended to hospitals, with the idea that entities were trying to benefit financially from the pandemic. One man reflected on this when he said,

> I don't know if it's true that hospitals when they have a death are attribut[ing] it to COVID because they get a financial benefit to calling it COVID if a 100 year old person [is] very compromised. I know of one case, an old person given three months to live died at the three month mark and they called it COVID!

The idea that medical entities were not to be trusted was related to population control and profit. Many discussed how people would suffer and die as the result of a government not caring about the general public. This also crossed political boundaries; however, the reason was not the same. People who self-reported right-leaning political affiliations were more concerned about powerful political entities who wanted to control the population via mandates. One man reflected on his view of the pandemic in similar ways to other interviewees. He said:

> It's control. Definitely control. The elite doesn't care about economics because they've already got their money. You are going to think I'm a conspiracy theorist, but you ever see The Hunger Games? I think that's what they want. You got extremely poor and extremely rich. Two classes. That's what you mostly have in every other socialist or communist country is you are either extremely rich or extremely poor. There is no middle class and I know this because I travel the world I know it because I see it. I see it in Mexico. I see it in South America. I mean Cuba is obviously that way. Either you are rich, and you work for the

government or you're super poor and you don't. That's what the left is after right here. Unfortunately, they have millions, tens of millions of sheep that *believe* that this is the best way and not how America started, and it will be how America ends.

In the center and on the left, however, people tended to worry more that government greed and accumulation would result in harm.

The [pandemic] response is very much in the same vein of putting economics and individualism over the care of our weakest in our community which is why you know it's no wonder why the most vulnerable communities have the highest rate of the disease.

In later interviews, people began to show increased similarities across political divides in mistrust of economic inequality and big business. Our later interviews occurred during the rollout of the first and second waves of the vaccine, and many people on both sides expressed concern about the rapidity of the development of the vaccine and conflicting or emerging data on efficacy. In Fig 2 we visually represent these areas of confluence and divergence between logics of blame.

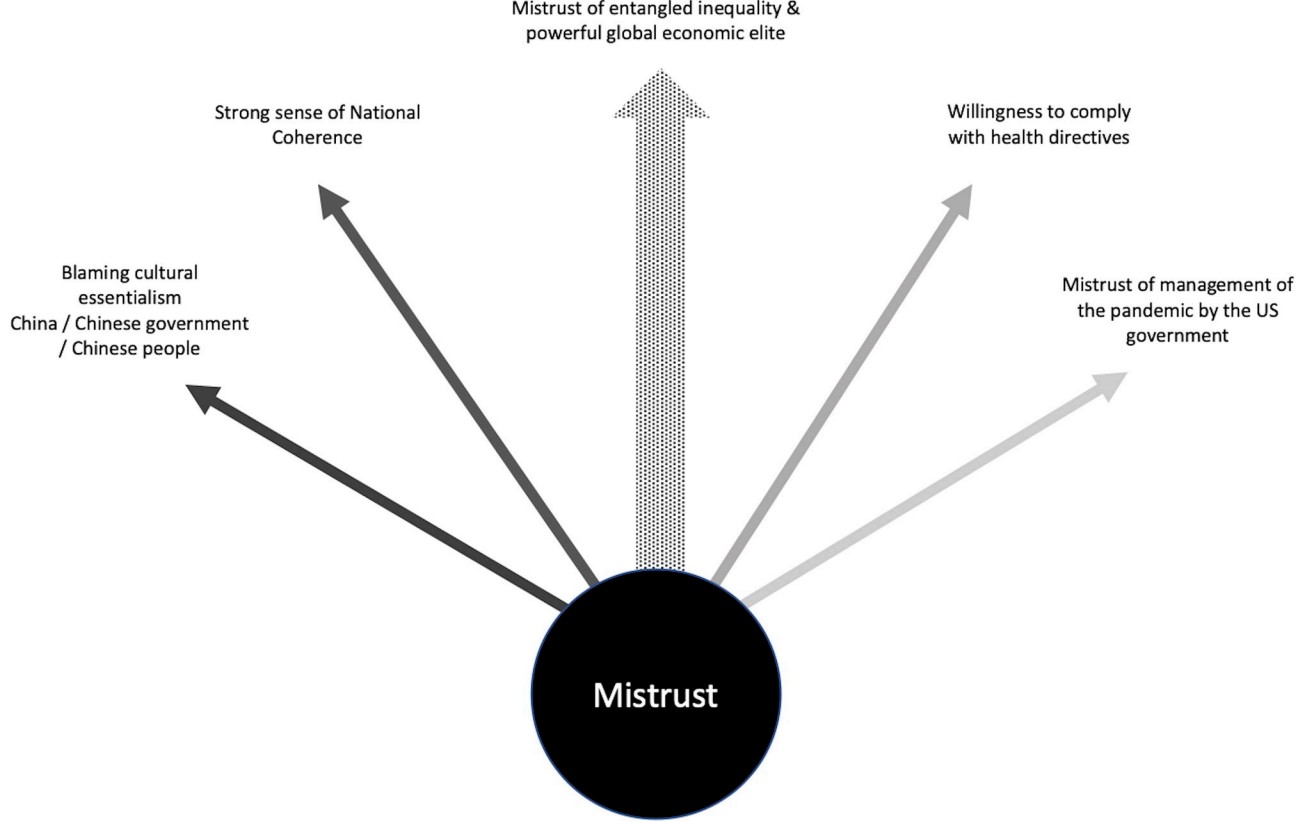

**Fig 2. Confluence and divergence.**

## Strengths and limitations

The strengths of this study include analysis of early and ongoing mixed methods data on the pandemic. Data collection began upon ethics approval in the first weeks of the declaration by the World Health Organization that COVID-19 was a pandemic. The design of the research allowed for measurement of immediate impacts and ongoing consistent themes.

Limitations of the use of these data include the inability to replicate real-time reactions to the pandemic. Salient themes that hold up throughout data collection are strong enough according to the methods design of the project that they could be replicated; however, details of the population level response to the beginning of the pandemic continue to change. Another limitation here is the focus on the United States. The larger study includes survey data from eight other countries; however, we only drew on the United States data for this article. Further research can continue to investigate political ideologies and health perceptions in other national settings. Finally, there are opportunities to continue analysis of these findings. The team plans to continue data collection and analysis to continue adding to the increasingly robust literature on COVID-19.

## Discussion

Overall mistrust and blame proliferate within perceptions of the virus and political actions related to the containment of contagion. Several points related to the similarities and differences of where blame falls across political lines are of note. First, the lack of trust and differences in SONC across political lines is significant and unusual. Multiple groups mistrust the federal government and have the sense that something is occurring at the federal level that is not in the best interest of the population of the country. The "what" and "how" of the problem, however, are different.

Racism and blame have long accompanied pandemics [36, 37]. Cultural essentialism is the practice of viewing "culture" as a static and exclusive descriptive category used to develop perceptions about other people based on small and often untrue interpretations of culture. Cultural essentialist perspectives have been used to undergird racism in multiple ways including associations between the non-human world and human behavior. Associating Chinese "culture" with eating bats is an example of this type of racism. Current pandemic responses reflect a long history of xenophobic blame and a climate of anti-Asian racism that has resulted in over two thousand reported hate crimes against Asian and Pacific Islander people across the United States [38, 39]. This scapegoating has occurred in everyday acts of hatred and also in the headlines and political speeches of public figures through phrases like "the Wuhan Coronavirus" and other tropes linking contagion with people of Asian descent [40]. This blame occurs more overtly among those on the political right than it does on the left; however, it is not absent from left voters who also often discussed questions of cultural essentialism as queries rather than objects of blame. Racism is present in social responses to the pandemic as well as healthcare systems that rely on the idea of privatized competitive markets and unequal healthcare systems [9].

The recognition of a rapidly growing class divide with a pinnacle of a small and powerful elite also seems to be a locus of anxiety across the political spectrum. This anxiety erodes believability and threatens the efficacy of public health interventions, setting the stage for conspiracy theories and causing a rift in understanding of public health directives and guidelines. While these data indicate higher trust in the economy on the right, lower trust of the WHO, CDC, and media due to ideas that they are controlled by a maleficent elite indicate agreement in popular and partisan beliefs, mistrust, and blame. And, though beliefs about a common good vary across partisan lines, there is shared support for COVID-19 economic relief [30]. It

is also within a common space across divides that perhaps the most eye-opening finding occurs. In a nation that is currently caught in the manifestations of late capitalism, which rests upon neoliberal governance, people are aware and afraid of the mechanisms of unequal accumulation. On both sides there is mistrust that there are people and entities in a position to accumulate wealth to the detriment of the human population. In this research, no matter where people stood on a political spectrum in the United States, they often discussed control by powerful entities and inequality as threats to life and health. It was in where they placed the blame that these perspectives diverged.

### Contributions to the literature

This research contributes directly to understanding pandemic response and the relationship between political and economic ideologies on health beliefs and possibly behaviors. These findings also demonstrate the benefits of an interdisciplinary use of medical social science methods and theories. Medical anthropologists and medical sociologists have the training and tools to delve into human behavior through systematic and mixed methods research and to make use of these methods for crisis response. Exploring and documenting ideologies and policy landscapes that contribute to inequality, racism, and other forms of injustice utilizes these disciplinary foundations and challenges researchers to create practical solutions. Literatures on critical medical anthropology and salutogenesis contribute to and benefit from frequent and rigorous use in an increasingly connected global system rife with present and future challenges to health and life.

### Conclusion

While the country remains deeply divided along political lines, there appear to be some areas of similar mistrust of other people based on ideologies of culture, economic systems, political leaders, and a global elite. Overall, across political differences, our respondents voiced their anxiety about inequality in terms of the outsized power of elites. It is within this economic and political moment that the need for deep and enduring systemic changes are revealed.

In these data we find that people across the political spectrum blame different agents for the country's challenges yet share a common link between logical flows: that there are powerful agents who accumulate financial and political power and may not have everyone's best interest at heart. This type of scapegoating, fear, and blame can potentially create the groundwork for political oppression. We suggest that this fear and blame may become an important focus of research and pandemic response through shifting the lens to an evaluation of the impacts of economic accumulation and the destruction of the safety net on the country as a whole.

### Implications

Given these findings, we believe that it is crucial to look beneath the surface of conspiracy theories to see that there is a real reaction to growing inequality and late capitalism. This may be at the root of understanding hesitancy and refusal to follow health directives. Increasing profit of medical manufacturing and a medical model that prioritizes profit over wellness is not a mystery to people living in the United States, who have endured navigating complicated health insurance plans, opaque and shifting plans for the rollout of vaccines, and awareness of inequality across populations. Attempting to improve adherence to public health directives and acceptance of vaccines and other medications will only be successful if we are willing to look carefully at a growing economic divide and a healthcare system that relies on profit over health and survival.

## Goals

1. Focus on critical solutions for the **reduction of extreme accumulation** and the **breakup of monopoly power** in healthcare. Political ideologies related to economic decision-making in the United States are divided; however, there is increasing bipartisan agreement that there is a need to reduce monopoly power. Reduction of extreme profit earning and power in this sector would increase public trust in vaccines and other crisis response measures

   a. This could potentially be achieved through **restructuring of healthcare systems** by redrawing the ecosystems in which healthcare entities operate. Healthcare systems designed on non-profit or similar models whereby profits for vaccine development and distribution is designated for public health and support as a public good could reduce mistrust and fear over extreme accumulation by a global elite. This is vital for future pandemic response.

2. **Transparency in health policy**. The proliferation of mistrust in government and corporate entities across political ideologies points to a need for transparent policies and procedures for international and global health. The Sunshine Act, for example, requires that "detailed information about payments and other 'transfers of value' worth over $10 from manufacturers of drugs, medical devices and biologics to physicians and teaching hospitals be made available to the public." [41] Other policies that require transparency will increase public trust in health directives and potentially increase vaccine uptake.

3. **Improve discussion of vaccine hesitancy**. Much public health messaging currently focuses on reducing vaccine hesitancy and increasing the uptake of vaccines in the United States. While this is certainly a goal for curbing the pandemic, there is a need for open discussions about why people fear vaccines and healthcare in general and steps necessary to change. This could potentially take the focus off of individual resistance and instead raise important questions about systemic inequalities that could be effective in improving trust and public health adherence.

4. **Prioritize research and education** on links between xenophobia/white supremacy and crisis response. Although our awareness is heightened in the United States today about organized white supremacist groups, xenophobic and racist views are also pervasive among those who do not participate in violent or well-organized hate groups. The kind of xenophobic blame that we saw in our research represents a severe threat to public health and must be addressed as such.

## Supporting information

**S1 Appendix. Interview guide.**
(DOCX)

**S2 Appendix. Questionnaire.**
(DOCX)

## Acknowledgments

The authors gratefully acknowledge the time of all people who participated through interviews and questionnaires, the CoRecovered COVID-19 group, and our international partnering researchers. We also thank Dr. Shifra Sagy for her leadership in salutogensis and vision for the

international collaboration. We extend our gratitude to the editor and anonymous reviewers of this manuscript whose time and attention improved this work and encouraged the inclusion of recommendations. The authors also acknowledge use of analysis software resources through the Southwest Health Equity Research Collaborative at Northern Arizona University (U54MD012388).

## Author Contributions

**Conceptualization:** Lisa J. Hardy, Eric Otenyo.

**Data curation:** Lisa J. Hardy, Adi Mana, Leah Mundell, Moran Neuman.

**Formal analysis:** Lisa J. Hardy, Adi Mana, Leah Mundell, Moran Neuman.

**Funding acquisition:** Lisa J. Hardy, Leah Mundell.

**Investigation:** Lisa J. Hardy, Adi Mana, Leah Mundell, Moran Neuman.

**Methodology:** Lisa J. Hardy, Adi Mana, Leah Mundell, Moran Neuman.

**Project administration:** Lisa J. Hardy, Leah Mundell, Sharón Benheim.

**Resources:** Lisa J. Hardy.

**Software:** Lisa J. Hardy.

**Supervision:** Lisa J. Hardy, Leah Mundell.

**Validation:** Lisa J. Hardy.

**Visualization:** Lisa J. Hardy, Sharón Benheim, Eric Otenyo.

**Writing – original draft:** Lisa J. Hardy.

**Writing – review & editing:** Lisa J. Hardy, Adi Mana, Leah Mundell, Moran Neuman, Sharón Benheim, Eric Otenyo.

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
