## [Decision Letter · Decision Letter 0]

10 May 2021

PONE-D-21-10928

Who is to blame for COVID-19? Examining politicized fear and health behavior through a mixed methods study in the United States

PLOS ONE

Dear Dr. Hardy,

Thank you for submitting your manuscript to PLOS ONE. After careful consideration, we feel that it has merit but does not fully meet PLOS ONE’s publication criteria as it currently stands. Therefore, we invite you to submit a revised version of the manuscript that addresses the points raised during the review process.

We look forward to receiving your revised manuscript.

Kind regards,

Prof. Anat Gesser-Edelsburg, Ph.D.

Academic Editor

PLOS ONE

Journal Requirements:

3) PLOS' guidelines require that conclusions are presented in an appropriate fashion and are supported by the data (https://journals.plos.org/plosone/s/criteria-for-publication#loc-4); this also applies to information presented in the Introduction that should be supported by the available literature. To this effect, please ensure that your statements are adequately and directly supported by the literature and findings. Authors may discuss possible implications for their results as long as these are clearly identified as hypotheses/interpretations instead of conclusions.

4) Please include additional information regarding the interview guide used in the study and ensure that you have provided sufficient details that others could replicate the analyses. Please include a copy, in both the original language and English, as Supporting Information.

5)  We note that you have indicated that data from this study are available upon request. PLOS only allows data to be available upon request if there are legal or ethical restrictions on sharing data publicly. For more information on unacceptable data access restrictions, please see http://journals.plos.org/plosone/s/data-availability#loc-unacceptable-data-access-restrictions.

6) We noted in your submission details that a portion of your manuscript may have been presented or published elsewhere. [Some of the quantitative results appear in another manuscript however they are used for a completely different interpretation and analysis. The only similarity is the standard description of the instruments we used. ] Please clarify whether this [conference proceeding or publication] was peer-reviewed and formally published. If this work was previously peer-reviewed and published, in the cover letter please provide the reason that this work does not constitute dual publication and should be included in the current manuscript.

7) Please ensure that you refer to Figure 2 in your text as, if accepted, production will need this reference to link the reader to the figure.

8) We note you have included a table to which you do not refer in the text of your manuscript. Please ensure that you refer to Table 4 in your text; if accepted, production will need this reference to link the reader to the Table.

Reviewers' comments:

Reviewer's Responses to Questions

**Comments to the Author**

1. Is the manuscript technically sound, and do the data support the conclusions?

Reviewer #1: Yes

Reviewer #2: Yes

2. Has the statistical analysis been performed appropriately and rigorously? 

Reviewer #1: Yes

Reviewer #2: Yes

3. Have the authors made all data underlying the findings in their manuscript fully available?

Reviewer #1: Yes

Reviewer #2: Yes

4. Is the manuscript presented in an intelligible fashion and written in standard English?

Reviewer #1: Yes

Reviewer #2: Yes

5. Review Comments to the Author

Reviewer #1: This is a very interesting scientific investigation on issues with important public health implications.

Some observations and comments are provided below.

Line 117 - Define salutogenesis

Line 224 'The team triangulated themes through this mixed methods approach' - The authors are encouraged to elaborate further on this as there are different types of triangulation (eg Denzin 1978). Explain also, eg provide examples of the quantitative and qualitative data convergence; or explain the significance of triangulation in this study.

METHODS

- Figure 1: Mixed methods design, and on line 305. Is this a new framework developed by the authors? If it is guided by prior work/ other authors, provide a reference.

- Line 238 used a "demographic screener to reduce sampling bias"; please elaborate or provide an example of this.

- Line 280 to 281 "The first related to health behaviors of mask wearing, distancing, and hand washing on a 7-point Likert scale (1= totally agree, 7= totally disagree)." This sounds like a single question was used to ask about three very distinct behaviors. If this was indeed the case, were each of these behaviors examined further in the qualitative component?

- Participant recruitment & use of the Prolific service. Approximately what percentage was from personal networks/ social media, and what was the proportion from Prolific?

- Line 313 "We found a high level of consistency between coders". To provide support to this statement, include either inter-coder reliability figures, or Explain what measures were taken to reduce subjectivity in coding.

- Line 343 - 344 "The data of 22 participants who defined themselves as far right and 88 far left, were deleted from the analysis". Why was this done? Was there a comparison of the results, if these N=110 were included in the analysis?

RESULTS

- Table 4 is a good visual depiction of the results. Can something similar be done for xxx

- The second part of the Results sections where participants were quoted verbatim, was very rich.

- Survey data results - In addition to correlational analysis, did the authors consider regression analysis to analyze relationships between the key variables of interest?

- Further investigation into the quantitative data seems possible. If the authors opted to focus primarily on the qualitative results, then this should be explained/ justified.

DISCUSSION

- In this section, explain what are the Implications / contributions of this research to the body of literature.

- How might the authors link their study findings to concepts of salutogenesis, Critical Medical Anthropology (CMA), and/or mixed methods methodology ?

- Discuss some implications to practitioners, policy makers, public health organizations, and/or government ? What can some of these stakeholders gain from your study's findings?

- Include a section where Limitations of this study are presented.

Reviewer #2: The manuscript is well written and is transparent. Has an interesting, engaging, and relevant topic. However, there is one concern about removing candidates that identify as far-right and far-left from the survey (denoted at 343 and 344). As of my conclusion, removing them would constrain an objective survey. I might be mistaken and there was a significant reason for removing them. It is just my concern about the objectivity of the survey that's all.

6. PLOS authors have the option to publish the peer review history of their article (what does this mean?). If published, this will include your full peer review and any attached files.

Reviewer #1: No

Reviewer #2: No

---

## [Author Response · Author response to Decision Letter 0]

7 Jun 2021

Reviewer comment

Author response

Please review your reference list to ensure that it is complete and correct. If you have cited papers that have been retracted, please include the rationale for doing so in the manuscript text or remove these references and replace them with relevant current references. Any changes to the reference list should be mentioned in the rebuttal letter that accompanies your revised manuscript. If you need to cite a retracted article, indicate the article’s retracted status in the References list and also include a citation and full reference for the retraction notice.

Done. We checked all references and have not cited retracted articles. We updated references where necessary.

 Done.

PLOS' guidelines require that conclusions are presented in an appropriate fashion and are supported by the data (https://journals.plos.org/plosone/s/criteria-for-publication#loc-4); this also applies to information presented in the Introduction that should be supported by the available literature. To this effect, please ensure that your statements are adequately and directly supported by the literature and findings. Authors may discuss possible implications for their results as long as these are clearly identified as hypotheses/interpretations instead of conclusions.

 Thank you for this thoughtful point. We revised the manuscript to ensure that we were not drawing unsupported conclusions. We also combined this with the reviewer comment that we could include implications and suggestions. We added a new section to explore these ideas and clarify there that these are only suggested ideas and not conclusions drawn directly from these data. This also helped us to refine the introduction. Our new section appears after the conclusion. 

Please include additional information regarding the interview guide used in the study and ensure that you have provided sufficient details that others could replicate the analyses. Please include a copy, in both the original language and English, as Supporting Information.

 We added the interview guide to the manuscript and supporting materials and linked it in the document. 

We also added the questionnaire is supporting material. 

Have the authors made all data underlying the findings in their manuscript fully available?

 We added our interview guide and questionnaire instruments as supporting materials. 

We are unable to share raw data because we do not have approval for this use of data through our Institutional Review Board. Interviews are long and in-depth. We are unable to de-identify them to the extent that they can be uploaded as supporting materials. We do include quotes throughout our published materials and information on number of interviews and protocols throughout the manuscript. In our response to the editor, we also include documentation of a discussion with research regulatory directors at our university documenting our inability to use a repository for these data. We plan to set up all future studies with the correct permissions and consent for adding data to a repository. 

We noted in your submission details that a portion of your manuscript may have been presented or published elsewhere. [Some of the quantitative results appear in another manuscript however they are used for a completely different interpretation and analysis. The only similarity is the standard description of the instruments we used.] Please clarify whether this [conference proceeding or publication] was peer-reviewed and formally published. If this work was previously peer-reviewed and published, in the cover letter please provide the reason that this work does not constitute dual publication and should be included in the current manuscript.

 We addressed this in the cover letter.

Please ensure that you refer to Figure 2 in your text as, if accepted, production will need this reference to link the reader to the figure.

 Corrected. 

We note you have included a table to which you do not refer in the text of your manuscript. Please ensure that you refer to Table 4 in your text; if accepted, production will need this reference to link the reader to the Table. 

 We analyzed data according to reviewer requests and changed the tables. We checked them to ensure formatting and correct headings in the text. 

Line 117 - Define salutogenesis

 Thank you. We defined salutogenesis and added some additional language in this section for clarity. 

Line 224 'The team triangulated themes through this mixed methods approach' - The authors are encouraged to elaborate further on this as there are different types of triangulation (eg Denzin 1978). Explain also, eg provide examples of the quantitative and qualitative data convergence; or explain the significance of triangulation in this study.

 Thank you for this question. We strengthened this section by including a clear definition of the types of triangulation we employed. We also cited additional sources including Denzin here. We think this is a substantial improvement to our description of the research process.

Figure 1: Mixed methods design, and on line 305. Is this a new framework developed by the authors? If it is guided by prior work/ other authors, provide a reference.

 Thank you for noting this. We did create this model. Our new text in this section cites other work on mixed methods and triangulation more thoroughly and clarifies that we created the design depicted in figure 1. 

Line 238 used a "demographic screener to reduce sampling bias"; please elaborate or provide an example of this.

 We clarified this process with this language: “the screening instrument allowed us to monitor who was included and select targeted volunteers from the available pool when necessary. When we noted that we had interviewed significantly more women for example, we selected men and gender non-conforming volunteers for interviews. Our goal was not to have a fully representative sample for the number of interviews but to ensure the reduction of an extremely limited sample.”

Line 280 to 281 "The first related to health behaviors of mask wearing, distancing, and hand washing on a 7-point Likert scale (1= totally agree, 7= totally disagree)." This sounds like a single question was used to ask about three very distinct behaviors. If this was indeed the case, were each of these behaviors examined further in the qualitative component?

 We improved this section and explained that these were different questions. In interviews we investigated general themes on health behaviors as described by participants and we did not ask deductive questions about different behaviors. We do have those data in the survey, however. 

Participant recruitment & use of the Prolific service. Approximately what percentage was from personal networks/ social media, and what was the proportion from Prolific?

 15%. We added this to the manuscript. 

Line 313 "We found a high level of consistency between coders". To provide support to this statement, include either inter-coder reliability figures, or Explain what measures were taken to reduce subjectivity in coding.

 Thank you for this important question. We returned to the literature to further define and elucidate this process. We added citations and wrote the following text to further explain the process we used: We deleted any codes that appeared in only one category. We found a high level of consistency between coders and, through our systematic approach of identifying and applying codes, removing outlying codes, and working together to discuss any discrepancies, we achieved intercoder consistency, a proven strategy for qualitative analysis [citations]. This check for intercoder consistency was enhanced by the use of three coders who participated as researchers on the project and had extensive methods training and had little interaction with one another prior to coding interviews in part due to the remote functioning of a pandemic project whereby interviews were conducted over the phone. This group had not discussed interview data at length or shared insights on findings prior to coding the interviews and only entered a more collaborative analysis phase once researchers decided on agreed upon themes and codes. 

Line 343 - 344 "The data of 22 participants who defined themselves as far right and 88 far left, were deleted from the analysis". Why was this done? Was there a comparison of the results, if these N=110 were included in the analysis? AND However, there is one concern about removing candidates that identify as far-right and far-left from the survey (denoted at 343 and 344). As of my conclusion, removing them would constrain an objective survey. I might be mistaken and there was a significant reason for removing them. It is just my concern about the objectivity of the survey that's all.

 Yes. This is an excellent point. We re-analyzed these data using all categories and made changes throughout the manuscript. Interesting (to us) was the fact that the researchers who analyzed these data were outside of the United States. Once we discussed this together we realized that this would have been a more useful decision in other political climates. Your comment here allowed us to improve that section of our data and employ triangulation here as well with our international perspectives. 

Survey data results - In addition to correlational analysis, did the authors consider regression analysis to analyze relationships between the key variables of interest?

Further investigation into the quantitative data seems possible. If the authors opted to focus primarily on the qualitative results, then this should be explained/ justified. Yes. Researchers improved this section by adding analysis. We also mention in the new section on limitations that we will continue to collect and analyze data further for future work. We agree that there are opportunities to continue and deepen analysis with available data on this and other themes. 

In this section, explain what are the Implications / contributions of this research to the body of literature.

Discuss some implications to practitioners, policy makers, public health organizations, and/or government? What can some of these stakeholders gain from your study's findings?

 Thank you for these excellent ideas. We added a section at the end of the manuscript. This also helped us to sharpen other sections. 

How might the authors link their study findings to concepts of salutogenesis, Critical Medical Anthropology (CMA), and/or mixed methods methodology ? We added a short section on contributions to these literatures. We hope that this addresses this comment and will continue to edit if this reviewer would like additional material in this area. 

Include a section where Limitations of this study are presented.

 Added.

 Done.

---

## [Decision Letter · Decision Letter 1]

30 Jul 2021

Who is to blame for COVID-19? Examining politicized fear and health behavior through a mixed methods study in the United States

PONE-D-21-10928R1

Dear Dr. Hardy,

We’re pleased to inform you that your manuscript has been judged scientifically suitable for publication and will be formally accepted for publication once it meets all outstanding technical requirements.

Kind regards,

Prof. Anat Gesser-Edelsburg, Ph.D.

Academic Editor

PLOS ONE

Additional Editor Comments (optional):

Reviewers' comments:

Reviewer's Responses to Questions

**Comments to the Author**

1. If the authors have adequately addressed your comments raised in a previous round of review and you feel that this manuscript is now acceptable for publication, you may indicate that here to bypass the “Comments to the Author” section, enter your conflict of interest statement in the “Confidential to Editor” section, and submit your "Accept" recommendation.

Reviewer #2: All comments have been addressed

2. Is the manuscript technically sound, and do the data support the conclusions?

Reviewer #2: Yes

3. Has the statistical analysis been performed appropriately and rigorously? 

Reviewer #2: Yes

4. Have the authors made all data underlying the findings in their manuscript fully available?

Reviewer #2: Yes

5. Is the manuscript presented in an intelligible fashion and written in standard English?

Reviewer #2: Yes

6. Review Comments to the Author

Reviewer #2: (No Response)

7. PLOS authors have the option to publish the peer review history of their article (what does this mean?). If published, this will include your full peer review and any attached files.

Reviewer #2: No

---

## [Editor Report · Acceptance letter]

9 Aug 2021

PONE-D-21-10928R1 

Who is to blame for COVID-19? Examining politicized fear and health behavior through a mixed methods study in the United States 

Dear Dr. Hardy:

I'm pleased to inform you that your manuscript has been deemed suitable for publication in PLOS ONE. Congratulations! Your manuscript is now with our production department. 

Kind regards, 

on behalf of

Prof. Anat Gesser-Edelsburg 

Academic Editor

PLOS ONE